# Quality of Life Assessment in Intestinal Stoma Patients in the Saudi Population: A Cross-Sectional Study

Reem Awad Alharbi [1], Nadeem Ahmad [1,*], Fatemah Yasser Alhedaithy [2], Majdoleen Dakhil N. Alnajim [1], Naima Waheed [3], Aisha A. Alessa [1], Banan A. Khedr [1] and Marriyam A. Aleissa [1]

[1] College of Medicine, Princess Nourah Bint Abdulrahman University, King Abdullah bin Abdulaziz University Hospital, Airport Road, King Khalid International Airport, Riyadh 11564, Saudi Arabia; reem.awad.alharbi@gmail.com (R.A.A.)

[2] Sulaiman Al-Habib Medical Group, College of Medicine, Princess Nourah Bint Abdulrahman University, Riyadh 11671, Saudi Arabia

[3] Department of Surgery, Alfaisal University, Riyadh 11533, Saudi Arabia; nwaheed@alfaisal.edu.sa

* Correspondence: naakhan@pnu.edu.sa; Tel.: +966-531413328

**Abstract:** Background: A stoma poses numerous physical, social, and psychological challenges and interferes with some religious practices, thus potentially negatively affecting the quality of life. In the contemporary era of stoma care, the study sought to assess this impact in a population with distinctive sociocultural characteristics. Methods: A modified City of Hope Quality of Life ostomy questionnaire was used to survey patients with intestinal stomas. The scoring was dichotomous on a 0 to 10 scale, where 0–3 indicated severe impact, 4–6 moderate, and 7–10 minimum. Statistical analysis involved Student's *t*-test, one-way ANOVA, Spearman's correlation, and multivariate linear regression. Results: There were 108 patients, with 59 males and 49 females. The mean age was 40.8 years. The overall quality of life score was 6, for the social domain 7, the physical domain 6, the psychological domain 5, and the spiritual domain 6. The stoma's impact on the quality of life was severe in 2%, moderate in 61%, and minimal in 37% of patients. Young patients, women, and those with benign diseases or without a job had low scores. Furthermore, 90% of patients had difficulty performing religious activities. For the regression analysis, life quality predictors were dietary, religious, pouch and stoma site issues, leak, odor, diarrhea or constipation, depression, anxiety, and future and disease concerns. Conclusions: Despite advances in stoma care, stoma patients had multiple impediments to their life quality. These were mainly psychological, but the physical and religious ones were also significant. A holistic approach to managing stoma patients is thus needed to help them have fulfilling lives.

**Keywords:** intestinal stoma; quality of life; stoma patients

## 1. Introduction

The intestinal stoma is a surgically created opening in the bowel for diverting bowel contents into a pouch for expulsion from the body. Even though contemporary surgical care makes it possible to maintain the integrity of the bowel, there are still circumstances where stoma creation is necessary. A stoma can save the lives of patients with severe sepsis and trauma, reduce the morbidity and mortality of the primary surgical procedure, and might be the definitive treatment for intractable fecal incontinence. Nonetheless, living with a stoma necessitates modifying one's lifestyle, as well as mental adjustment [1].

A stoma may be temporary or permanent, and connected to the small or large intestine, referred to as ileostomies or colostomies, respectively. It is performed most frequently for colorectal cancer and inflammatory bowel disease [2].

Since 1771, when the first documented ostomy was performed [3], there has been continuous research and innovation to lessen the negative impact of stomas on patients' quality of life (QoL), leading to improvements in surgical technique, appliances, and overall stoma care. However, it is unclear to what extent these developments improved the Muslim

patients' QoL, particularly those with religious obligations [4,5]. Therefore, the present study was conducted to determine the QoL in these patients and to identify areas to focus on while devising stoma coping strategies.

## 2. Materials and Methods

This was a survey-based study conducted at the College of Medicine, Princess Nourah Bint Abdulrahman University, in Riyadh. Included patients were Saudi nationals who had an ileostomy or colostomy for more than three weeks, were at least 18 years old, and were able to provide informed consent. Those with severe disabilities, such as requiring a wheelchair, being bedridden, having advanced cancer, or having a preexisting mental disorder, were excluded. The patients were approached through a community stoma care nurse. The data were collected from August 2020 to April 2022.

We utilized a modified version of Marcia Grant's City of Hope QoL ostomy questionnaire, which included specific questions regarding the type of stoma appliances, local culture and religious practices, and stoma care facilities. Its Arabic translation, to meet the needs of locals, was double-checked for its authenticity by two Arabic-native speakers independently. A pilot study with Cronbach's alpha of 0.6 and Pearson's interclass correlation coefficient of 0.3 validated the modified version. The questionnaire consisted of 63 items, covering socio-demographics and stoma-related issues (21 items), along with the physical (17 items), social (12 items), psychological (8 items), and spiritual (5 items) dimensions of quality of life.

This was cross-sectional study with systematic sampling and targeted a minimum sample size of 100. Patients self-administered their responses after giving informed consent and receiving the online questionnaire. The University's Institutional Review Board (IRB) approved the study in a letter dated 22 June 2020, with IRB number 20-0245.

Each predictor variable was ranked by a dichotomous scale for the ease of responders and interpretation. Responses with a positive impact on QoL were scored as 1 and with a negative as 0. Contrary responses were reverse-coded. For comparison with other studies, the total scores were converted to a 0 to 10 scale. As in other studies [6], a QoL score of 0 to 3 implied severe impact, 4 to 6 moderate impact, and 7 to 10 minimal impacts. IBM SPSS version 25 was used for the data analysis. Normality of data was checked using a Q–Q plot. Numerical variables were presented as means with standard deviation and range, whereas categorical variables were presented as absolute numbers and percentages. Student's *t*-test and one-way ANOVA were used to compare mean Qol scores between different groups and Spearman's correlation was used to determine the relationship between QoL score and stoma-related factors. Using a multivariable linear regression model, significant QOL score predictors were identified.

## 3. Results

The response rate was 46%; 108 out of 234 patients who received the questionnaire answered it. Their mean age was $40.8 \pm 13.7$ years (range 20–77), with 59 (54.6%) males and 49 (45.4%) females. The majority (86%) held a high school diploma or a postsecondary degree. The proportions of married and employed patients were similar before and after the stoma. However, the marital status of some individuals changed, with 6% divorcing after the stoma. The distribution of ileostomies and colostomies and whether these were temporary or permanent was nearly identical. After Crohn's disease, the second most common underlying cause was colorectal cancer. Many respondents preferred a particular brand of stoma bag—Hollister being the most popular. Table 1 shows the detailed sociodemographic characteristics of the patients.

**Table 1.** Comparison of the sociodemographic characteristics of patients with their quality of life.

| Variables | | Frequency | QOL Score | *p* Value ˆ | Variables | | Frequency | QOL Score | *p* Value ˆ |
|---|---|---|---|---|---|---|---|---|---|
| Age Groups | <60 | 96 | 6.06 ± 1 | 0.74 | Stoma Status | Temporary | 40 (37%) | 6 ± 1 | 0.1 ** |
| | 60 or more | 12 | 6.17 ± 1.3 | | | Permanent | 42 (38.9%) | 6 ± 1.1 | |
| Gender | Male | 59 (54.6%) | 6.28 ± 1 | 0.03 | | Not known | 26 (24.1%) | 5.9 ± 0.9 | |
| | Female | 49 (45.4%) | 5.82 ± 1 | | Underlying disease | Crohn' disease | 32 (29.6%) | 6.3 ± 1.1 | 0.3 ** |
| BMI | <35 | 10 | 5.83 ± 0.6 | 0.4 * | | Cancer | 29 (26.9%) | 5.6 ± 1 | |
| | 35 or more | 98 | 6.09 ± 1.1 | | | Ulcerative Colitis | 8 (7.4%) | 6.4 ± 0.8 | |
| Residence | East | 22 (20.4%) | 6 ± 1 | 0.1 ** | | Intestinal Obstruction | 10 (9.3%) | 5.9 ± 1.1 | |
| | West | 21 (19.4%) | 5.9 ± 1 | | | Fecal Incontinence | 11 (10.2%) | 5.8 ± 0.8 | |
| | North | 16 (14.8%) | 5.8 ± 0.9 | | | Others | 18 (16.7%) | 6 ± 1.1 | |
| | South | 13 (12%) | 5.7 ± 1 | | Disease groups | Cancer | 29 (27%) | 6.2 ± 1.3 | 0.5 * |
| | Central | 36 (33.3%) | 6.1 ± 1.1 | | | Non cancer disorders | 79 (73%) | 6 ± 1 | |
| Stoma Duration | Up to 1 year | 34 (31.5%) | 5.9 ± 1 | 0.8 * | Pouch Brand | Hollister | 36 (33.3%) | 6.2 ± 1 | 0.3 ** |
| | >1 year | 74 (68.5%) | 6 ± 1 | | | Coloplast | 14 (13%) | 6.1 ± 1.4 | |
| Education | College | 61 (56.5%) | 5.9 ± 1 | 0.6 ** | | Convetac | 4 (3.7%) | 5.2 ± 0.9 | |
| | Diploma | 18 (16.7%) | 6.1 ± 0.74 | | | Others | 54 (50%) | 5.9 ± 0.9 | |
| | Masters | 7 (6.5%) | 6.1 ± 1.2 | | Pouch Preference | Yes | 42 (38.9%) | 6 ± 1.2 | 0.8 * |
| | Secondary | 19 (17.6%) | 6 ± 1.3 | | | No | 66 (61.1%) | 6 ± 0.9 | |
| | Intermediate | 1 (0.9%) | 5.5 | | Time for diet adjustment | 6 months or less | 80 (74.1%) | 6.09 ± 1.1 | 0.7 * |
| | Illiterate | 2 (1.9%) | 6.4 ± 2 | | | >6 months | 28 (25.9%) | 6.01 ± 0.9 | |
| Marital status before stoma | Single | 48 (44%) | 5.8 ± 1.1 | 0.06 * | Time for stoma adjustment | 6 months or less | 86 (79.6%) | 6.06 ± 1 | 0.8 * |
| | Married | 60 (56%) | 6.2 ± 1 | | | >6 months | 22 (20.4%) | 6.11 ± 1.1 | |
| Marital after stoma | Single | 37 (41.7%) | 5.9 ± 1 | 0.2 * | Job before stoma | Yes | 60 (55.6%) | 5.9 ± 0.9 | 0.6 * |
| | Married | 63 (58.3%) | 6.2 ± 1 | | | No | 48 (44.4%) | 6 ± 1.2 | |
| Stoma Type | Colostomy | 41 (38%) | 6.2 ± 1.1 | 0.3 ** | Job after stoma | Yes | 60 (55.6%) | 6.10 ± 1 | 0.7 * |
| | Ileostomy | 45 (41.7%) | 5.9 ± 1.2 | | | No | 48 (44.4%) | 6.03 ± 1.1 | |

\* Student's *t*-test; \*\* One-way ANOVA. ˆ Significance level is 0.05.

The overall QoL score, and scores attained in the physical and spiritual domains were all 6 (Table 2). The psychological domain score was 5 and the social domain was 7 (all figures rounded off). As per QoL score criteria, the stoma's impact on QoL was severe in 2% (*n* = 2), moderate in 61% (*n* = 66), and mild in 37% (*n* = 40) of patients.

**Table 2.** Quality of Life's Domains and Categories' Scores.

| Domains | Scores | Categories | | |
|---|---|---|---|---|
| **QoL Score (Mean)** | **6.1 ± 1.1 (3–9)** | | | |
| Physical | 5.8 ± 1.2 (3–8) | Scores range | Impact on QoL | Frequency |
| Social | 6.8 ± 1.7 (2–10) | 0–3 | Severe | 2 (1.9%) |
| Psychological | 4.9 ± 2.1 (0–10) | 4–6 | Moderate | 66 (61.1%) |
| Spiritual | 6.2 ± 2.4 (0–10) | 7–10 | Mild | 40 (37%) |

Patients had a wide range of QoL scores based on their demographic characteristics, as shown in Table 1. For example, older patients had higher scores than younger ones, and the same was true for males versus females, married versus unmarried, and employed versus unemployed. Furthermore, patients with cancer had better scores than those with benign diseases, and those with colostomy scored better than those with an ileostomy.

Regarding religious activities, 90% of the patients felt restrictions in performing one or more of these—mainly the wudhu (ablution) keeping. Other affected activities were going to a mosque, saying regular prayers, or fasting, all of which resulted in lower QoL scores (Table 3).

**Table 3.** Quality of life scores based on religious factors.

| Religious Factors | No & Percent of Patients | QoL Score | *p* Value * |
|---|---|---|---|
| Restriction in performing any of religious activity | 97 (90%) | 6 ± 1.1 | 0.002 |
| No restriction in performing any of religious activity | 11 (10%) | 7.1 ± 0.8 | |
| Difficulty in visiting a Mosque | 46 (43%) | 5.7 ± 1 | <0.001 |
| No difficulty in visiting a Mosque | 62 (57%) | 6.4 ± 1 | |
| Difficulty in keeping wudhu | 62 (57%) | 5.8 ± 1.2 | <0.001 |
| No difficulty in keeping wudhu | 46 (43%) | 6.5 ± 0.9 | |
| Difficulty in keeping fast | 37 (34%) | 5.6 ± 1.1 | <0.001 |
| No difficulty in keeping fast | 71 (66%) | 6.4 ± 1 | |
| Difficulty in saying prayers regularly | 37 (34%) | 5.6 ± 1.2 | 0.001 |
| No difficulty in saying prayers regularly | 71 (66%) | 6.4 ± 1 | |
| Difficulty in reciting Quran | 22 (20%) | 5.8 ± 1.4 | 0.19 |
| No difficulty in reciting Quran | 86 (80%) | 6.2 ± 1 | |

* Student's *t*-test.

In the correlation analysis, shown in Table 4, there was a statistically significant correlation between QoL and all five religious' factors, with seven of the eight psychological factors, ten of the twelve social factors, and eight of the eighteen physical factors.

**Table 4.** Correlations between patients' factors and quality of life score.

| Patients' Factors | Patients' No & Percent | Correlation Coefficient [#] | *p* Value | Patients' Factors | Patients' No & Percent | Correlation Coefficient [#] | *p* Value |
|---|---|---|---|---|---|---|---|
| | | | Physical Dimension | | | | |
| Dietary restrictions: Any such restriction | 88 (81%) | 0.28 ** | 0.003 | Stoma site problems | 53 (49%) | 0.44 ** | <0.001 |
| | | | | Clothing changed | 78 (72%) | 0.25 ** | 0.009 |
| Carbonated drinks | 55 (51%) | 0.23 * | 0.02 | Low physical strength | 43 (40%) | 0.12 | 0.19 |
| Dairy products | 41 (38%) | 0.11 | 0.27 | Fatigue | 27 (25%) | 0.06 | 0.55 |
| Fruits | 24 (22%) | 0.02 | 0.86 | Aches & pains | 25 (23%) | 0.07 | 0.49 |
| Vegetables | 20 (19%) | 0.05 | 0.58 | Sleep affected | 83 (77%) | 0.07 | 0.48 |
| Leak | 69 (64%) | 0.18 | 0.06 | Travelling affected | 39 (36%) | 0.39 ** | <0.001 |
| Odor | 43 (40%) | 0.28 ** | 0.003 | ER visits for complications | 39 (36%) | 0.11 | 0.26 |
| Constipation | 12 (11%) | 0.20 * | 0.03 | Skin Allergy | 27 (25%) | 0.01 | 0.95 |
| Diarrhea | 19 (18%) | 0.17 | 0.08 | Skin irritation | 50 (46%) | 0.09 | 0.33 |

**Table 4.** *Cont.*

| Patients' Factors | Patients' No & Percent | Correlation Coefficient # | p Value | Patients' Factors | Patients' No & Percent | Correlation Coefficient # | p Value |
|---|---|---|---|---|---|---|---|
| | Social Dimension | | | | Psychological Dimension | | |
| Personal relations affected | 26 (24%) | 0.48 ** | <0.00 | Depression | 56 (51.9%) | 0.39 ** | <0.001 |
| Self-isolation | 24 (22%) | 0.44 ** | <0.00 | | | | |
| Family distress | 63 (58%) | 0.15 | 0.11 | Anxiety | 89 (82%) | 0.39 ** | <0.001 |
| Self-independence | 97 (90%) | 0.32 ** | 0.001 | Embarrassment | 35 (32%) | 0.15 | 0.13 |
| Self-management of stoma | 95 (88%) | 0.20 * | <0.03 | Concern for underlying disease | 81 (75%) | 0.30 ** | 0.001 |
| Financial burden | 37 (34%) | 0.13 | 0.19 | Concern for future life | 73 (68%) | 0.31 ** | 0.001 |
| Friends & Family support | 87 (81%) | 0.37 ** | <0.001 | Optimism | 96 (89%) | 0.27 ** | 0.004 |
| Social activities affected | 60 (56%) | 0.49 ** | <0.001 | Satisfaction with stoma life | 84 (78%) | 0.34 ** | <0.001 |
| Stoma care briefing by health professional | 79 (73%) | 0.21 * | 0.03 | Pouch problems concerns | 71 (66%) | 0.18 | 0.06 |
| Stoma care education by health professional | 78 (72%) | 0.30 ** | 0.002 | | Spiritual Dimension | | |
| Sex life affected | 69 (64%) | 0.41 ** | <0.001 | Religious activities restrictions: Any such restriction | 97 (90%) | 0.3 * | 0.002 |
| Social services support at home | 80 (74%) | 0.3 ** | 0.002 | Visiting Mosque | 46 (43%) | 0.33 ** | 0.001 |
| | | | | Keeping Wudhu | 62 (57%) | 0.31 | 0.001 |
| | | | | Keeping Fast | 37 (34%) | 0.36 ** | <0.001 |
| | | | | Saying Regular prayers | 37 (34%) | 0.31 ** | 0.001 |
| | | | | Reciting Quran | 22 (20%) | 0.04 | 0.67 |

# Spearman's Correlation. * Correlation is significant at a 0.05 level. ** Correlation is significant at a 0.01 level.

The significant predictors of QoL identified by the multivariate linear regression model were dietary and religious, pouch and stoma site issues, leak, odor, constipation or diarrhea, depression, anxiety, and concerns regarding the future and underlying disease (Table 5). Other important factors that determined QoL included the effect of the stoma on personal relationships; self-independence; ability to self-manage the stoma; and the availability of family, friends, or social services support.

**Table 5.** Quality of life predictor evaluation using multivariate linear regression #.

| Predictors | Regression Coefficients (Beta) | p Value | Predictors | Regression Coefficients (Beta) | p Value |
|---|---|---|---|---|---|
| Dietary restrictions: | | | Personal relations | 0.43 | <0.001 |
| Carbonated drinks | 0.37 | <0.001 | | | |
| Dairy products | 0.17 | 0.03 | Social Activities | 0.24 | 0.008 |
| Fruits | 0.23 | 0.02 | Self-isolation | 0.16 | 0.14 |
| Vegetables | 0.27 | 0.005 | Family support | 0.03 | 0.72 |
| Leak | 0.40 | <0.001 | Sex life | 0.25 | 0.003 |
| Odor | 0.23 | 0.005 | Self-management of stoma | 0.03 | 0.82 |
| Diarrhea | 0.30 | 0.002 | Self-independence | 0.23 | 0.07 |

**Table 5.** *Cont.*

| Predictors | Regression Coefficients (Beta) | *p* Value | Predictors | Regression Coefficients (Beta) | *p* Value |
|---|---|---|---|---|---|
| Constipation | 0.26 | 0.04 | Stoma care briefing by healthcare provider | 0.18 | 0.04 |
| Stoma location | 0.43 | <0.001 | | | |
| Clothing changed | 0.39 | <0.001 | Home social support | 0.18 | 0.07 |
| Travelling interfered | 0.06 | 0.52 | Depression | 0.36 | <0.001 |
| Religious restrictions: | | | Anxiety | 0.27 | 0.01 |
| Going to Mosque | 0.33 | <0.001 | Disease concerns | 0.29 | 0.001 |
| Keeping Wudhu | 0.25 | 0.003 | Future concerns | 0.22 | 0.01 |
| Keeping Fasting | 0.28 | <0.001 | Pouch problems concerns | 0.14 | 0.13 |
| Saying Prayers | 0.28 | <0.001 | Satisfaction with stoma life | 0.37 | <0.001 |
| Reciting Quran | 0.27 | 0.004 | Optimism | 0.09 | 0.47 |

[#] R Square = 0.92, F value = 39, ANOVA ≤ 0.001.

## 4. Discussion

Stoma care has evolved significantly, becoming a specialty, along with advances in surgical techniques and effluent collection devices [7]. As a result, stoma patients are now believed to be better able to manage their stomas and carry out their daily lives, including participating in their religious practices. Still, patients from various cultural, socioreligious, and economic backgrounds are expected to adjust to their stomas differently [8]. With this perspective, the current study is valuable as it highlights these variations in the context of the Saudi population, which has unique sociocultural characteristics and religious fervor.

The questionnaire used in the present study measured various aspects of QoL as specified by WHO [9,10]. We modified the original version to meet the objectives of our study and to facilitate the patients to respond clearly and straightforwardly taking the minimum time and simplifying the scoring.

We found varied QoL scores in people with different sociodemographic groups. For instance, consistent with the results of other studies, younger patients had significantly lower scores than older ones—possibly because the former have more emotional and future concerns [11]. Women also had somewhat lower QoL scores than men, which could be due to their worries about their altered body images and constraints in doing household tasks [8]. Other studies, however, contradict this, as women were found to be more likely to seek social support and participate in social activities to improve their quality of life [7]. Obesity has been linked to stoma retraction and improper pouch fitting [12]. However, we could not find any correlation between patient body mass index and QoL. A person's marital status also had little impact on QoL, but the post-stoma 6% divorce rate, which had equal gender distribution, is socially disturbing. The disruption to sexual life could have contributed to it.

A steady increase in the prevalence of inflammatory bowel disease in Saudi Arabia explains why it was ranked as the most prevalent underlying cause of stoma in our patients. In contrast, in most published studies, it was colorectal cancer [13].

Given the high prevalence of colorectal cancer in Saudi Arabia, cancer was the second leading cause of stomas among our patients [14]. Cancer patients had slightly better QoL scores than those with benign conditions, possibly because of their better mental preparedness to cope with the consequences of a life-threatening illness. Patients with colostomies had higher scores than those with ileostomies, obviously because colostomies were easier to manage than ileostomies. The evidence that employment improves QoL, and joblessness lowers it was validated by our regression analysis [15].

The overall QoL score in our study population was 6, which was lower than what studies done in other countries have reported. For example, in Brazil it was 6.2, Australia

6.9, Netherlands 7.1, India 7.5, and the United States 7.3 [3,10,16]. None of our patients scored 10/10, which would have indicated unimpaired QoL. However, according to the score-based criteria, most of our patients had a moderate or minimal impact on QoL.

Stoma patients are likely to face several physical problems because of their changed anatomy. The most troublesome and negatively impacting QoL is a poor stoma site, precipitated by obesity, emergency surgery, cancer, and improper technique [17,18]. A poorly placed stoma increases the risk of leakage, an ill-fitting pouch, and parastomal complications, and has been associated with sexual dysfunction, insomnia, weight loss, and depression [19,20]. It was also a significant negative factor in our patients' QoL.

Leak and offensive odor are common issues in stoma patients. These are frustrating as these interfere with activities at home, work, sports, and in social and religious settings. In the Ostomy Life Study, 91% of patients expressed concern about leakage [20]. Our study endorsed this. Steps to prevent or mitigate them are worth exploring.

About 75% of stoma patients suffer from parastomal skin complications—the majority due to leakage and adhesives [20]. These were more common in patients who had ileostomies rather than colostomies. Nevertheless, their incidence in our patients was lower (49%) than reported.

Patients' QoL can also be negatively impacted if they develop complications such as prolapse, hernia, and obstruction. Their incidence is higher in patients treated in substandard medical facilities [21]. These could prompt patients to visit the hospital in an emergency, further adding to their stress. A third of our patients required such a visit; attention to the surgical technique safeguards against these complications [18].

Despite their widespread occurrence and contrary to common belief, physical issues were not the most crucial determinant of low QoL. This could be because the patients were mentally more prepared to deal with them. On the contrary, psychological factors bothered patients the most and thus adversely affected their QoL. Some other studies also showed similar findings [22]. Among these factors, anxiety was the most common, followed by concerns about the underlying disease, pouch problems, and depression. Included in the list by other studies were altered body image, hopelessness, stigmatization, isolation, and loneliness [16,23]. Sadly, most healthcare providers ignore these vital psychological concerns and only concentrate on physical issues [22]. However, recently, the awareness about them has increased, and many counter strategies have been proposed [22,24].

The spiritual well-being of an individual was a primary focus of our study, as were considered it to be an essential component of QoL [25]. A stoma could make it difficult for Muslims to maintain the cleanliness required to fulfill their religious obligations, necessitating frequent going to the bathroom, which might be inconvenient. Patients could be further discouraged from praying, fasting, and performing Hajj due to concerns about leaks and odor [26].

As all the participants in our study were devout Muslims, our survey questions covered all the rituals typically observed by Muslims. A very high proportion of responders reporting difficulties in performing one or more religious activities indicates that modern stoma appliances are not foul-proof and might need additional innovations to ensure patients' comfort and convenience. Another significant issue is the need for appropriate religious counseling of the patients, as proposed by many authors [7,27,28].

Stoma patients often struggle with social dysfunction and sexual issues, lowering their QoL. Disinterest, avoiding social activities, restricted travel, strained personal relationships, loneliness, and a tendency toward isolation are of significant concern. These factors make adaptation and stoma management even more challenging. The fact that 44% of our patients exhibited social disturbances and our inference that sexual dysfunction negatively impacted QoL match with the evidence [22,23]. This negative effect can be lessened by giving patients access to pertinent information and training and by providing efficient community health services [7].

Most of our patients (91%) were able to self-manage their stomas and live independently—thanks to the quality of care they received and the support of their family and friends. Their financial difficulties were also minimal due to the country's effective national health program.

Only Muslims of Saudi descent were included in our study. Thus, surveying people with diverse racial and ethnic backgrounds will be intriguing. We also used traditional statistical methods to analyze our data. However, future surveys of this nature should consider a more recent approach that employs multivariate statistical methods such as cluster analysis, factor analysis, and principal component analysis [29].

## 5. Conclusions

Stoma patients have multiple impediments to their having a good quality of life. Contrary to the common perception, the most crucial issue psychological factors. In addition, however, physical, and religious issues also remained significant, despite advances in stoma care. Thus, there is a need for a comprehensive, patient-centered approach with the integration of medical resources to address all patients' concerns and to mitigate the negative effects of stoma fallouts.

**Author Contributions:** R.A.A. writing and editing the manuscript; N.A. conceptualization, data analysis, and writing and editing of the manuscript; F.Y.A. conceptualization, data collection and interpretation, writing and editing of the manuscript; M.D.N.A. conceptualization, data collection and interpretation, and writing and editing of the manuscript; N.W. data collection and processing, and writing and editing of the manuscript; A.A.A. collection and processing of references; B.A.K. collection of references and writing of the manuscript; M.A.A. collection of references and writing of the manuscript. All authors have read and agreed to the published version of the manuscript.

**Funding:** The study didn't receive any funding.

**Institutional Review Board Statement:** The study was conducted in accordance with the Declaration of Helsinki and approved by Princess Nourah Bint Abdulrahman University's Institutional Review Board (IRB) in a letter dated 22 June 2020, with IRB number 20-0245.

**Informed Consent Statement:** Informed consent was obtained from all subjects involved in the study.

**Data Availability Statement:** Original data of the study are available on the repository, figshare. Link: https://figshare.com/s/fe26cf6fc9c70b1eaebd (accessed on 4 January 2023).

**Acknowledgments:** We would like to extend our sincere gratitude to Areej M. Alqatifi, an enterostomal therapist working at King Fahd Specialist Hospital in Dammam, Saudi Arabia, for the invaluable assistance she provided with the collection of data.

**Conflicts of Interest:** The authors declare no conflict of interest.

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
