# Peer review of "Quality of Life Assessment in Intestinal Stoma Patients in the Saudi Population: A Cross-Sectional Study"

_gastroent, doi:10.3390/gastroent14030022_

Round 1

Reviewer 1 Report

The study is interesting and original, it is dedicated to an important medical problem. The approach of the authors offers the use of a specific questionnaire applied to certain number of participants. The selection of descriptors is well designed and the classical statistics used for data interpretation is correct. The reference list is well selected but I would like to offer at least a new citation dealing with a novel approach to medical questionnaire treatment and interpretation.

My major comment and suggestion (rather for future similar studies than to changes in the present scheme of work) is related to application of multivariate statistical data mining and respective interpretation as offered in the recommended for citation publication:

M. Nedyalkova, J. Romanova, L. Naneva, V. Simeonov. Phys. Sci. Rev., https/doi.org/10.1515/psr -2021-0158 The authors will be convinced of the advantages of the multivariate methods (machine learning) to reveal specific relationships between objects (participants) and variables (selected descriptors for quality of life after stoma).

The English language is correct, minor corrections are necessary

Author Response

  1. The reference suggested has been added in the discussion and in the reference list.
  2. Necessary corrections of English language mistakes have been done in the text.  

Reviewer 2 Report

This paper reported the quality of life assessment in intestinal stoma patients in the Saudi population with a well-designed City of Hope Quality of Life ostomy questionnaire and the statistical analysis. The study revealed that stoma patients had multiple impediments to their life quality despite advances in stoma care and a holistic approach to managing stoma patients was needed to help them have fulfilling lives. This topic fits the scope of this journal and may benefit the development of professional patient care especially for these intestinal stoma patients. The manuscript is well-written and the results can support the conclusions. The following key issues are required to be addressed before its publication on Gastroenterol Insights.

1. The previous reported studies are required to be simply introduced in the Introduction section.

2. More detailed questionnaire information are required for the research reproduction purpose.

3. The figure (columns or curves) illustrations are required to show the difference for a better readability of this manuscript.

4. The limitations of this research are required to be discussed in the Discussion section.

The English language is fine.

Author Response

  1. References to previous studies have been included in the introduction section (Ref 4 & 5).
  2. The questionnaire and its validation  has been elaborated in the Methods section.

  3. Results have been well summarized in the table so figures ommited to avoid duplication.

  4. A paragraph has been added at the discussion's end to cover it.

Reviewer 3 Report

Dear Authors

The study from a scientific point of view seems to be well done and presents good results, from where to derive valid conclusions. This study holds significant importance for public health. Intestinal stomas, which involve surgically creating an opening in the bowel for diverting bowel contents, play a crucial role in treating and managing various conditions. While contemporary surgical advancements have improved bowel integrity, stoma creation remains necessary in certain circumstances. Stomas can save lives by addressing severe sepsis, trauma, and intractable fecal incontinence, while also reducing morbidity and mortality associated with primary surgical procedures. However, living with a stoma requires significant lifestyle modifications and mental adjustment. Understanding the experiences, challenges, and needs of individuals with stomas can inform healthcare strategies and support services, ultimately improving the overall well-being and quality of life for this population.

This cross sectional study was conducted at the College of Medicine, Princess Noura bint Abdulrahman University, in Riyadh. The participants were Saudi nationals with ileostomy or colostomy for over three weeks, aged 18 or older, and capable of providing informed consent. Individuals with severe disabilities or preexisting mental disorders were excluded. The patients were contacted through a community stoma care nurse, and data collection took place from August 2020 to April 2022.

Minor Corrections:

The reference list needs some attention according to journal instructions.

Author Response

Necessary corrections have been done in the references. 

Reviewer 4 Report

The submitted article is of very good quality and fits your journal with its focus, I have only slight comments.

The key words are repeated in the title.

Please add information about the location where the data collection was taking place in the material and methodology section.

Please also provide the estimated size of this population.

Author Response

Key words have been  revised as advised.